

# Comprehensive transcriptome analysis provides new insights into nutritional strategies and phylogenetic relationships of chrysophytes

Daniela Beisser[1], Nadine Graupner[2,3], Christina Bock[2,3], Sabina Wodniok[2,3], Lars Grossmann[2,3], Matthijs Vos[4], Bernd Sures[5], Sven Rahmann[1,*] and Jens Boenigk[2,3,*]

[1] Genome Informatics, University of Duisburg-Essen, Essen, Germany
[2] Biodiversity, University of Duisburg-Essen, Essen, Germany
[3] Centre for Water and Environmental Research (ZWU), University of Duisburg-Essen, Essen, Germany
[4] Theoretical and Applied Biodiversity, Ruhr-University Bochum, Bochum, Germany
[5] Aquatic Ecology, University of Duisburg-Essen, Essen, Germany
[*] These authors contributed equally to this work.

Corresponding author
Daniela Beisser,
daniela.beisser@uni-due.de

## ABSTRACT

**Background.** Chrysophytes are protist model species in ecology and ecophysiology and important grazers of bacteria-sized microorganisms and primary producers. However, they have not yet been investigated in detail at the molecular level, and no genomic and only little transcriptomic information is available. Chrysophytes exhibit different trophic modes: while phototrophic chrysophytes perform only photosynthesis, mixotrophs can gain carbon from bacterial food as well as from photosynthesis, and heterotrophs solely feed on bacteria-sized microorganisms. Recent phylogenies and megasystematics demonstrate an immense complexity of eukaryotic diversity with numerous transitions between phototrophic and heterotrophic organisms. The question we aim to answer is how the diverse nutritional strategies, accompanied or brought about by a reduction of the plasmid and size reduction in heterotrophic strains, affect physiology and molecular processes.

**Results.** We sequenced the mRNA of 18 chrysophyte strains on the Illumina HiSeq platform and analysed the transcriptomes to determine relations between the trophic mode (mixotrophic vs. heterotrophic) and gene expression. We observed an enrichment of genes for photosynthesis, porphyrin and chlorophyll metabolism for phototrophic and mixotrophic strains that can perform photosynthesis. Genes involved in nutrient absorption, environmental information processing and various transporters (e.g., monosaccharide, peptide, lipid transporters) were present or highly expressed only in heterotrophic strains that have to sense, digest and absorb bacterial food. We furthermore present a transcriptome-based alignment-free phylogeny construction approach using transcripts assembled from short reads to determine the evolutionary relationships between the strains and the possible influence of nutritional strategies on the reconstructed phylogeny. We discuss the resulting phylogenies in comparison to those from established approaches based on ribosomal RNA and orthologous genes. Finally, we make functionally annotated reference transcriptomes of each

strain available to the community, significantly enhancing publicly available data on Chrysophyceae.

**Conclusions.** Our study is the first comprehensive transcriptomic characterisation of a diverse set of Chrysophyceaen strains. In addition, we showcase the possibility of inferring phylogenies from assembled transcriptomes using an alignment-free approach. The raw and functionally annotated data we provide will prove beneficial for further examination of the diversity within this taxon. Our molecular characterisation of different trophic modes presents a first such example.

## INTRODUCTION

Recent phylogenies and megasystematics demonstrate an immense complexity of eukaryotic diversity with numerous transitions between phototrophic and heterotrophic organisms (*Adl et al., 2012*; *Keeling, 2004*; *Boenigk, Wodniok & Glücksman, 2015*). While a primary endosymbiosis of a cyanobacterium into a eukaryotic host cell (thus originating eukaryotic photosynthesis) is considered to be a singular event (*Keeling, 2004*), except for the case of the cercozoan genus *Paulinella*, secondary and tertiary endosymbiosis, i.e., the acquisition of a eukaryotic algae by a eukaryotic host cell, occurred several times (*Keeling, 2004*; *Petersen et al., 2014*). The subsequent loss of pigmentation and of the phototrophic ability presumably occurred by far more often. The highest diversity of secondarily colourless lineages is currently attributed to the Stramenopiles, specifically the chrysophytes comprising both phototrophic and heterotrophic forms. The evolution of heterotrophs occurred presumably at least five to eight times independently within chrysophytes (classes Chrysophyceae Pascher 1914 and Synurophyceae Andersen 1987; *Kristiansen & Preisig (2001)*; *Andersen (2007)*). The chrysophytes are therefore particularly suited for addressing the evolution of colorless algae.

Due to a varying degree of loss of pigmentation and phototrophic ability, a wide range of different nutritional strategies is realized in chrysophytes (heterotrophic, mixotrophic, phototrophic). While phototrophic chrysophytes perform photosynthesis and heterotrophs solely feed on bacteria or small protists, mixotrophs can use a mix of different sources of energy and carbon through digestion of microorganisms and photosynthesis. Chrysophytes with different strategies typically co-exist in diverse habitats, but vary in performance under changing environmental conditions.

Chrysophytes have for decades served as protist model species in ecology and ecophysiology (*Montagnes et al., 2008*; *Pfandl, Posch & Boenigk, 2004*; *Rothhaupt, 1996b*; *Rothhaupt, 1996a*); they are among the most important grazers of bacteria-sized microorganisms (*Finlay & Esteban, 1998*) and, specifically in oligotrophic freshwaters, an important component of the primary producers (*Wolfe & Siver, 2013*). Nevertheless,
they have not yet been investigated in detail at a molecular level, and no genomic and only little transcriptomic information of related organisms (*Terrado et al., 2015*; *Keeling et al., 2014*; *Liu et al., 2016*) is available.

Evolutionary relationships between organisms are usually represented as phylogenetic trees which are often inferred from the gene sequences of orthologous genes (*Ciccarelli et al., 2006*; *Wu & Eisen, 2008*). Current knowledge on chrysophyte phylogeny is largely based on single gene analyses of the small subunit ribosomal RNA gene (SSU rDNA) (*Pfandl et al., 2009*; *Stoeck, Jost & Boenigk, 2008*; *Boenigk, 2008*; *Scoble & Cavalier-Smith, 2014*; *Bock et al., 2014*; *Grossmann et al., 2016*) as well as one multigene analysis (*Stoeck, Jost & Boenigk, 2008*). The increasing taxon sampling during the past years contributed to our current understanding of the chrysophyte phylogeny. The affiliation of taxa and strains to distinct orders within Chrysophyceae based on SSU rRNA gene sequence data has stabilized during the past years. Molecular data now support the position of scale-bearing phototrophic taxa (*Synura* spp. and *Mallomonas* spp.) as order Synurales within Chrysophyceae (*Scoble & Cavalier-Smith, 2014*; *Grossmann et al., 2016*). Apart from this phototrophic clade the orders Ochromonadales, Chromulinales, Hydrurales and Hibberdiales are consistently supported in SSU phylogenies. The unpigmented scale-bearing taxa formerly lumped within the genus *Paraphysomonas* have recently been revised and based on the evidence provided the two paraphysomonad families Paraphysomonadidae and Clathromonadidae also seem to be well supported and separated in SSU phylogenies (*Scoble & Cavalier-Smith, 2014*). However, the precise branching order of the major chrysomonad clades varies with algorithm and taxon sampling (*Scoble & Cavalier-Smith, 2014*; *Grossmann et al., 2016*; *Bock et al., 2014*). Current molecular phylogenetic analyses concentrate on few chrysophyte taxa such as the phototrophic genera *Synura* and *Mallomonas* (*Škaloud, Kristiansen & Škaloudová, 2013*; *Siver et al., 2015*) and the mixotrophic genus *Dinobryon* (*Bock et al., 2014*), as well as on mixotrophic and colourless single-celled taxa originally lumped into the genera *Paraphysomonas* (*Scoble & Cavalier-Smith, 2014*), *Spumella* (*Grossmann et al., 2016*) and *Ochromonas* (*Andersen (2007)* and RA Andersen, pers. comm., 2007). The fragmentary taxon coverage and a presumably early radiation of the Chrysophycea so far conceal the relation of chrysophyte orders and families. Similarly, intra-clade phylogenies are in many cases unsatisfactorily resolved. Again taxon coverage is an issue here. On top of that, the phylogenetic resolution of the SSU rRNA gene reaches its limits for analysis in particular of intrageneric and intraspecific diversity (*Boenigk et al., 2012*). Furthermore, in particular the findings of numerous colourless lineages within Chrysophyceae separated by mixotrophic lineages as indicated by SSU rRNA phylogenies heated the discussion on the suitability of single gene phylogenies and on the SSU rRNA gene as a gene to reflect the evolutionary history of chrysophytes. Even though the SSU rRNA gene is still considered to be the gold standard for molecular phylogenies in chrysophytes, the multiple evolution of colorless lineages within an algal taxon intensified the demand for multigene or genome-/transcriptome-scale analyses.

In recent years, several alternative approaches have been proposed to infer phylogenies based on properties of the whole genome, such as gene content, gene order, genome sequence similarity and nucleotide frequencies (*Reva & Tümmler, 2004*; *Coenye et al., 2005*;
*Delsuc, Brinkmann & Philippe, 2005*; *Snel, Huynen & Dutilh, 2005*; *Pride et al., 2006*; *Patil & McHardy, 2013*; *Chan & Ragan, 2013*; *Fan et al., 2015*). These approaches are less biased by any single locus, computationally cheap, and therefore ideal for the comparison of several large genomes. By using statistical properties of the genome, they are in most cases able to work on even incompletely assembled sequences and are less affected by misassemblies.

To our knowledge, alignment-free methodologies have not yet been applied to transcript sequences. Recent studies that used transcriptome data to infer phylogenies either use sequencing technologies which produce long reads such as Roche 454 (*Borner et al., 2014*), or short read sequences in combination with available reference genomes of the species (*Wen et al., 2013*). Usually, all transcripts that belong to a set of orthologous genes are used for a combined multiple sequence alignment (*Peters et al., 2014*), from which the trees are then built. However, certain transcripts might not be expressed (and hence not observed) under study conditions, which may significantly reduce the set of available genes with complete orthology information. Additionally, properly dealing with alternative transcripts of the same gene may be non-trivial. We therefore describe an alignment-free $k$-mer approach for assembled transcriptomes, apply it to the 18 chrysophyte RNA-seq datasets and discuss the resulting phylogenies in comparison to a gene-based approach.

In summary, this article makes three contributions.

1. On the data side, we provide a valuable dataset of RNA-seq data and functionally annotated assembled transcripts for 18 diverse Chrysophyceaen strains with different nutritional strategies.
2. On the analysis side, we assess the relations between trophic mode and gene content and expression differences at the metabolic pathway level.
3. On the methodological side, we discuss phylogenetic inference from assembled transcriptomes based on alignment-free $k$-mer methods.

The main objectives of our study are (1) to investigate relations between trophic mode and molecular processes at the transcriptome level and (2) to use abundantly available transcriptome data as an additional source for phylogeny reconstruction.

## MATERIALS AND METHODS

### Strain cultivation and sample preparation

All strains were grown at 15 °C in a light chamber with 75–100 µE illumination (1 E[instein] is defined as the energy in $6.022 \times 10^{23}$ photons) and a light:dark cycle of 16:8 h. Light intensities were adapted to conditions allowing for near maximum oxygen evolution but still below light saturation in order to avoid adverse effects (*Rottberger et al., 2013*). Due to different pH requirements of the investigated strains different media were used: Most heterotrophic strains were grown in inorganic basal medium (*Hahn et al., 2003*) with the addition of *Listonella pelagia* strain CB5 as food bacteria (*Hahn, 1997*), exceptions from this are mentioned separately below. The inorganic basal medium for the axenic strains was supplemented with 1 g/l of each of nutrient broth, soytone and yeast extract (NSY; *Hahn et al. (2003)*) in order to allow for heterotrophic growth. *Poteriospumella lacustris* strains JBM10, JBNZ41 and JBC07 as well as *Poterioochromonas malhamensis* strain DS were grown

axenically in the culture collection of the working group. For details on origin, isolation procedure and axenicity of the axenic strains, see *Boenigk & Stadler (2004)*; *Boenigk et al. (2004)*. *Dinobryon* strain LO226KS, *Synura* strain LO234KE and *Ochromonas/Spumella* strain LO244K-D were grown in DY-V medium (*Andersen, 2007*). *Dinobryon* strain FU22KAK, *Epipyxis* strain PR26KG and *Uroglena* strain WA34KE were grown in WC medium (*Guillard & Lorenzen, 1972*). We did not expect strong effects of the media on gene expression and tested for this during data analysis (see Results and Discussion).

Cells for RNA isolation were harvested by centrifugation at 3,000 g for 5–10 min at 20 °C. RNA extraction was carried out under sterile conditions using TRIzol (Life Technologies, Paisley, Scotland–protocol modified). Pellets were ground in liquid nitrogen and incubated for 15 min in TRIzol. Chloroform was added and the mixture was centrifuged to achieve separation of phases. The aqueous phase was transferred to a new reaction tube and RNA was precipitated using isopropanol (incubation for 1h at −20 °C and centrifugation). The RNA pellet was washed three times in 75% ethanol and re-suspended in diethylpyrocarbonate (DEPC) water.

## Sequencing

Preparation of the complementary DNA (cDNA) library as well as sequencing was carried out using an Illumina HiSeq platform via a commercial service (Eurofins MWG GmbH, Ebersberg, Germany). An amplified short insert cDNA library (poly-A enriched mRNA) with an insert size of 150–400 base pairs (bp) was prepared per sample, individually indexed for sequencing on Illumina HiSeq 2000, sequenced using the paired-end module and then demultiplexed.

## Quality control and preprocessing of sequencing data

The quality control tool FastQC (v0.10.1; *Andrews (2012)*) was used to analyse the basepair quality distribution of the raw reads. Adapter sequences at the ends of the reads were removed using the preprocessing software Cutadapt (v1.3; *Martin (2011)*). Cutadapt was also used to trim bad quality bases with a quality score below 20 and discard reads with a length below 70 bp after trimming.

## Assembly and annotation

Clean reads were de-novo assembled to transcript sequences with Trinity (Release 2013-11-10, default parameters; *Grabherr et al. (2011)*) and Oases (v0.2.08; *Schulz et al. (2012)*) with different $k$-mer sizes and multiple-$k$-mer approaches using $k \in \{19, 21, 27, 35, 39, 43, 51, 57, 75\}$. For transcript quantification the clean reads were remapped on transcripts using the short read mapper Bowtie2 (v2.2.1, with parameters –all -X 800; *Langmead & Salzberg (2012)*) and counted with the transcript quantification tool eXpress (v1.3.1, with parameters: –fr-stranded –no-bias-correct; *Roberts & Pachter (2013)*). RAPSearch2 (v2.15, default parameters, but $\log_{10}(E\text{-value}) < -1$; *Zhao, Tang & Ye (2012)*) which uses six-frame translation and a reduced amino acid alphabet for rapid protein similarity search was used to assign transcripts to the best hit searching all genes in the Kyoto Encyclopedia of Genes and Genomes (KEGG) (Release 2014-06-23; *Kanehisa & Goto (2000)*). In this way the transcripts were annotated with KEGG Orthology IDs

(KO ID) and KEGG pathways. All analysis steps were performed using the workflow environment Snakemake (v3.2.1; *Köster & Rahmann (2012)*).

## Expression analysis

Transcript quantification was performed with the tool eXpress (v1.3.1; *Roberts & Pachter (2013)*) which resolves multimappings to estimate transcript abundances in multi-isoform genes. KEGG Orthology gene counts were summarized thereupon as the sum over the effective transcript counts. By this approach, the expression of all transcripts of one gene will be summarized to a common gene with the most conserved function. Potential paralogous genes with the same KEGG Orthology ID will be consolidated, too, potentially increasing gene counts for some genes. This yields a coarse view on expression, but a detailed analysis of novel transcripts and genes of unknown functions for all strains is outside the scope of this manuscript.

For differential expression analysis, the R package DESeq2 (v1.6.3; *Love, Huber & Anders (2014)*) was used. DESeq2 models the count data as negative binomial distributed, estimates the variance-mean dependence and tests for differential expression. For visualization the counts were variance stabilized, normalized for sample size and a principal component analysis (PCA) was performed with a corresponding plot of the first principal components using the R package vegan (v2.3-0; *Oksanen et al. (2015)*). Each axis reveals relations between groups of samples and data points. Samples and data points having high similarity with respect to this relation have similar coordinates in the plot. For reasons of clarity only the samples were depicted in the plots. The PCA was performed on the 500 genes with the highest variance.

The significantly differential genes were used subsequently in an enrichment analysis. The pathways were reduced to plausible metabolic pathways, removing pathways in the KEGG categories global and overview maps, human diseases and drug development. Own implementations were used to perform a hypergeometric test for each KEGG pathway and pathway visualisation. All mappings of genes to KEGG pathways and pathways with a significant enrichment were reported.

All figures in R were created using ggplot2 (v1.0.1; *Wickham (2009)*).

## Alignment-based phylogenetic inference

For the alignment-based phylogenetic inference, the KEGG database (*Kanehisa & Goto, 2000*) was used to find orthologous genes between all strains. Multiple sequence alignments were constructed with MAFFT (v7.164b, parameters: –maxiterate 1,000 – adjustdirectionaccurately –op 2 –globalpair; *Katoh & Standley (2013)*) for overlapping regions of the transcripts of all 18 strains. For each alignment the longest transcript was used. The alignments were manually checked and corrected with Jalview (v2.8.2b1; *Waterhouse et al. (2009)*) and concatenated thereupon to create one multigene alignment. Based on the multiple sequence alignment the phylogeny estimation was performed in R with the package phangorn (v1.99-12; *Schliep (2011)*). A model test was used to obtain the best substitution model. The general time-reversible model with gamma distribution and number of invariant sites (GTR+G+I) was the best fit for the data and used subsequently to estimate the maximum-likelihood phylogeny with a bootstrap analysis.

**Table 1 Species and strains of Chrysophyceae used in this study.** Shown are species, strain, nutrition type (trophic mode: hetero, heterotrophic; mixo, mixotrophic; photo, phototrophic; ax, axenically grown; a question mark '?' means that the trophic mode is under discussion), number of raw read pairs per sample, clean read pairs per sample after preprocessing, and percentage of clean read pairs per sample.

| Species | Strain | Trophy | Raw read pairs | Clean read pairs | % Clean |
|---|---|---|---|---|---|
| *Spumella vulgaris* | 199hm | hetero | 13,899,445 | 12,843,053 | 92.40 |
| *Cornospumella fuschlensis* | A-R4-D6 | hetero | 14,575,684 | 13,452,316 | 92.29 |
| *Acrispumella msimbaziensis* | JBAF33 | hetero | 15,061,239 | 13,296,427 | 88.28 |
| *Pedospumella sinomuralis* | JBCS23 | hetero | 14,432,210 | 13,216,644 | 91.58 |
| *Spumella bureschii* | JBL14 | hetero | 15,494,585 | 14,447,147 | 93.24 |
| *Apoikiospumella mondseeiensis* | JBM08 | hetero | 15,431,398 | 11,578,366 | 75.03 |
| *Poteriospumella lacustris* | JBM10 | hetero ax | 19,351,386 | 17,745,650 | 91.70 |
| *Pedospumella encystans* | JBMS11 | hetero | 14,150,502 | 13,156,769 | 92.98 |
| *Spumella lacusvadosi* | JBNZ39 | hetero | 16,079,979 | 14,219,198 | 88.43 |
| *Ochromonas* or *Spumella sp.* | LO244K-D | hetero? | 18,662,530 | 8,414,623 | 45.09 |
| *Dinobryon sp.* | FU22KAK | mixo | 17,828,441 | 8,627,549 | 48.39 |
| *Poteriospumella lacustris* | JBC07 | hetero ax | 13,841,037 | 12,796,750 | 92.46 |
| *Poteriospumella lacustris* | JBNZ41 | hetero ax | 18,752,714 | 17,332,930 | 92.43 |
| *Dinobryon sp.* | LO226KS | mixo | 19,572,799 | 17,917,338 | 91.54 |
| *Synura sp.* | LO234KE | photo | 13,310,559 | 11,256,835 | 84.57 |
| *Poterioochromonas malhamensis* | DS | mixo ax | 15,822,091 | 14,917,952 | 94.29 |
| *Epipyxis sp.* | PR26KG | mixo | 17,915,043 | 17,284,316 | 96.48 |
| *Uroglena sp.* | WA34KE | mixo | 22,107,199 | 18,106,322 | 81.90 |

For the SSU phylogeny the sequences were edited with DNADragon (v1.5.2; *Hepperle (2012)*) and aligned in BioEdit Sequence Alignment Editor (v7.1.3.0; *Hall (1999)*) using the ClustalW algorithm (default settings) and manual editing by eye. The SSU alignment follows a compilation of sequences (provided by J.M. Scoble) covering all known lineages of Chrysophyceae. 17 of the 18 investigated strains were added using *Sellaphora blackfordensis* and *Nannochloropsis granulata* as outgroups. The maximum-likelihood phylogenetic tree and corresponding robustness measures (bootstrap analyses with 1,000 replicates) were inferred with Treefinder (*Jobb, von Haeseler & Strimmer, 2004*) using GTR+I+G as model of evolution.

All trees were visualized with FigTree (v1.4.2; *Rambaut (2012)*).

## RESULTS AND DISCUSSION

A total of 18 chrysophyte transcriptomes of the genera *Acrispumella, Apoikiospumella, Cornospumella, Dinobryon, Epipyxis, Ochromonas, Pedospumella, Poterioochromonas, Poteriospumella, Spumella, Synura* and *Uroglena* were paired-end sequenced on the Illumina HiSeq2000 platform which yielded between 13 and 22 million read pairs per sample (Table 1). The reads were subsequently cleaned to remove adapter sequences and low quality reads, resulting in 8–18 million read pairs (45.09%–96.48%, mean 85.17%, median 91.64%).

The data is provided as a public resource at the European Nucleotide Archive (ENA) database, study accession PRJEB13662 (*Beisser et al., 2016*). We provide both raw read sequences and assembled transcript sequences (see below).

## Assembly of transcripts

The cleaned reads were de-novo assembled to transcripts with the software Trinity and then quantified at the transcript level for expression analysis. Previous attempts to assemble with the software Oases resulted in shorter contigs with poorer alignment results when searching against the Uniprot and NCBI database.

The statistics for the assembled transcriptomes are shown in Fig. 1, sorted by GC content of the strains, which incidentally also separates the trophic modes (Fig. 1A). Trinity outputs assembled *transcripts* and additionally groups them into *components* based on shared sequence content. Such a component is loosely referred to as a gene, the idea being that the contained transcripts are isoforms or variants of the same gene. The total number of bases in the transcriptomes (Fig. 1B) are in the range of 3,164,810 bp–51,472,244 bp. *Dinobryon* strain FU22KAK shows the lowest value. For this sample, many of the raw reads were removed during preprocessing due to insufficient quality values which might be one reason for its worse performance in the assembly process. The N50 length of an assembly (i.e., of a set of contigs of total length $L$) is the smallest length $N$ for which the set of contigs of length $\geq N$ contains at least $L/2$ nucleotides. The N50 values of our assemblies range between 404 and 1,566, with a mean value of 912 (Fig. 1C). The average number of transcripts lies at 36,637 with *Dinobryon* strain FU22KAK showing the lowest number of transcripts (8,275) and *Spumella bureschii* (JBL14) the highest (72,269). The number of components is on average 25,433 (Fig. 1D). Most components contain only a low number of transcripts. In particular, 71.26%–91.42% (median 84.52%) of the components have only one transcript. There are no sequenced reference genomes available for these Chrysophyceae strains, so their genome size and their number of genes is unknown. We may take the number of components (Fig. 1D) as a rough estimate for the number of genes. Similar sizes were found for four prymnesiophyte species, with a transcriptome size between 35.3 Mbp–52.0 Mbp and 30,986–56,193 contigs (*Koid et al., 2014*).

## Functional annotation of transcripts

Transcripts were assigned to a KEGG genes, orthologs (KO) and pathways. Figure 2 shows how many gene and pathway annotations were obtained. The number of pathway annotations exceeds the number of KO annotations since one KO may belong to more than one pathway. Despite the difficulty to map sequences of organisms with a low similarity to the KEGG reference genes, 44–86% and 10–32% of the sequences were successfully annotated to KEGG genes and orthologs. Due to the high evolutionary distance to the organisms present in KEGG, we expect to assign mainly the more conserved genes responsible for basic cellular functions and processes and not to assign the strain-specific genes. These transcripts remain without KEGG annotation since the sequence similarity to KEGG genes is too low or the genes are not present in the KEGG database.

By KO assignment, known functional information associated with a gene in a specific organism is transferred to the strains under consideration. Using this approach, we may lose information about recently duplicated genes (paralogs). We also counted the number of unique KOs and pathways hit by any transcript in each sample; see Fig. 2. The range of unique pathways is between 225 and 265. The number of unique KOs and pathways

 

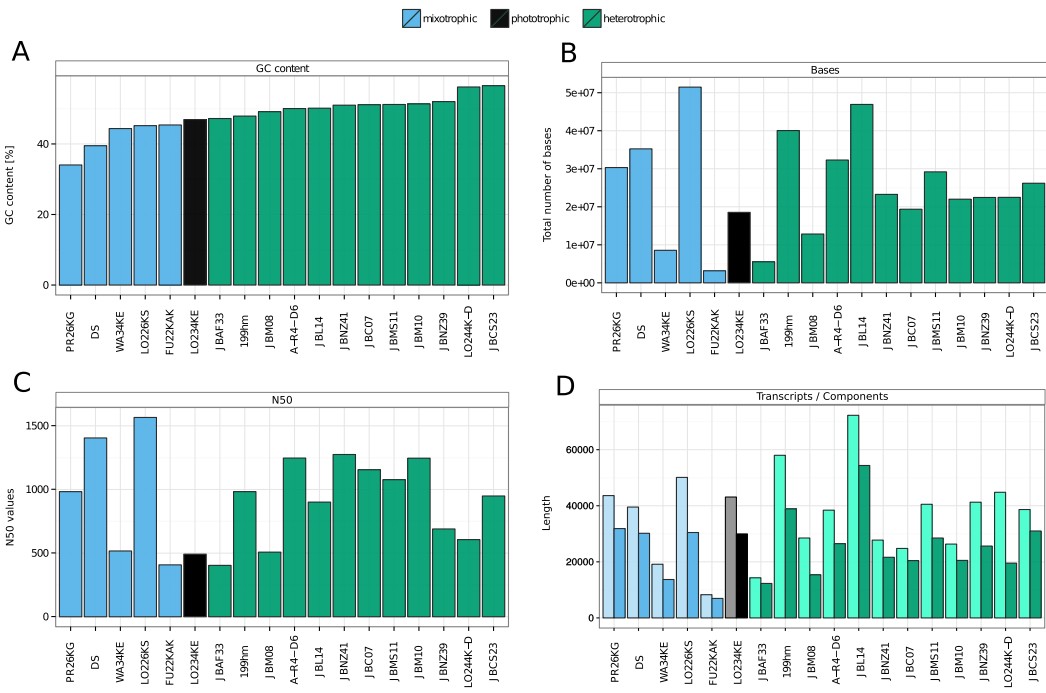

**Figure 1** **Statistics for transcriptomes assembled with Trinity.** (A) shows the GC content of each strain. Since the GC content separates the trophic modes, the other panels were also sorted by GC content. (B) shows the estimated transcriptome sizes (total number of bases in transcriptome). (C) shows N50 value of each strain (contig length such that half of the transcriptome is in contigs longer than this length). (D) shows the number of assembled transcripts (light colour) and components (approximately genes; dark colour) for each sample.

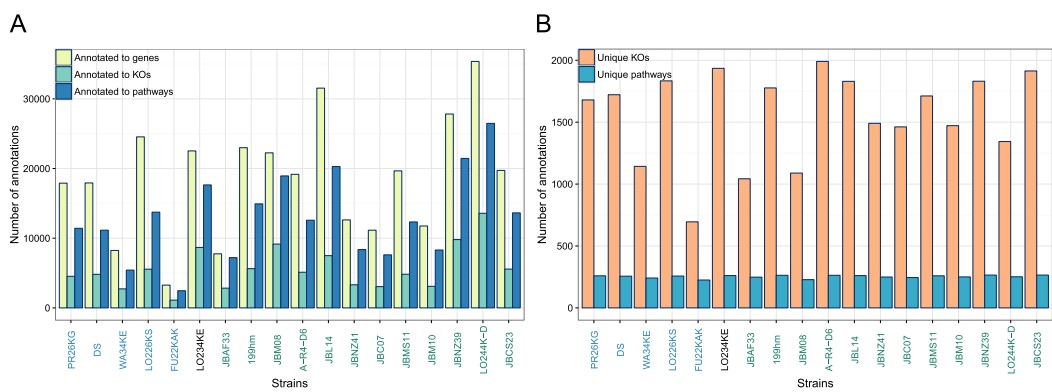

**Figure 2** **Number of annotations to KEGG genes, KEGG Orthology IDs (KOs) and to KEGG pathways from all transcripts of each sample (A) and the number of unique KOs and KEGG pathways for each sample (B).** Strains on the *x*-axis are sorted and coloured according to Fig. 1.

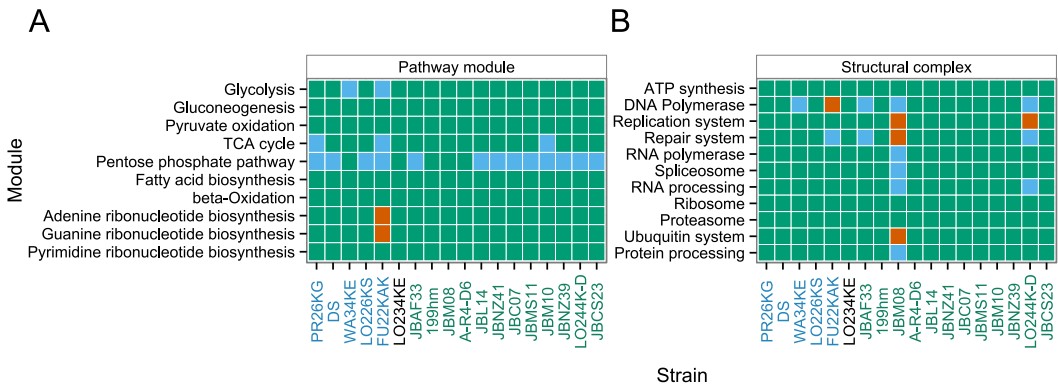

**Figure 3** **Completeness of KEGG essential modules.** (A) shows pathway modules, (B) shows structural complexes. Modules are considered operational if all enzymes necessary for the reaction steps or proteins constituting a complex are present. Pathway modules are coloured in green if at most one enzyme is missing, in blue if more than one enzyme is missing, but the central module is complete (e.g., complete module: M00001 Glycolysis (Embden-Meyerhof pathway); central module: M00002 Glycolysis, core module involving three-carbon compounds) and in red if more than one enzyme and the core module are missing. Structural complexes consist of several modules, e.g., ATP synthesis consists of 22 modules. Structural complexes are coloured in green if the majority of the modules is functional, in blue if less then half of them are present and in red if the structural complex is missing. Strains on the $x$-axis are sorted and coloured according to Fig. 1.

is similar in all strains and indicates a similar coverage of the KEGG reference pathways (Fig. 2B). Using the number of unique KOs as a proxy for functionally conserved genes present in the strains, we obtain on average 1,696 genes.

The completeness of the sequenced transcriptomes was assessed by testing the operationality of essential KEGG modules (see Fig. 3). These modules are a collection of manually defined functional units which require that all enzymes necessary for the reaction steps or proteins constituting a complex are present. We selected essential modules from primary metabolism and structural complexes that are required for the functioning of the cell, including central carbohydrate metabolism, fatty acid metabolism, nucleotide metabolism, ATP synthesis, DNA polymerase, replication system, repair system, RNA polymerase, spliceosome, RNA processing, ribosome, proteasome, ubiquitin system and protein processing. All species, except FU22KAK and JBM08, cover the essential modules. These observations mostly imply good coverage and completeness of the assembled transcriptomes. The *Dinobryon* strain FU22KAK lacks part of the central carbohydrate metabolism and nucleotide metabolism, which hints to a quality issue which was already described in the last section. *Apoikiospumella mondseeiensis* (strain JBM08) contains complete gene sets for the pathway modules, but misses some of the gene sets necessary for the structural complexes, which is also evident from the highly expressed pathways (see next section) and likely caused by the transcriptional state of the cell.

## General molecular characterisation
In general, the most actively transcribed pathways comprise ribosome synthesis as well as protein processing and biosynthesis of several amino acids (Fig. 4). Since ribosome maintenance as well as general transcription and translation are essential for protein

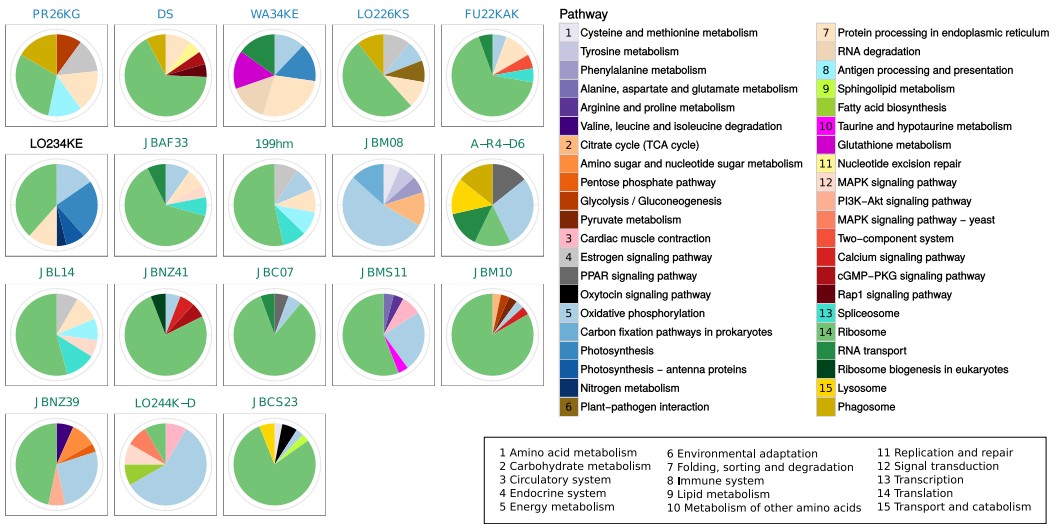

**Figure 4  Most highly expressed pathways per sample, such that contained genes explain 50% of the total expression.** Pathways are coloured and grouped according to their KEGG hierarchy II displayed below the pathway legend, e.g., all blue colours, number 5, belong to energy metabolism. Samples names are sorted and coloured according to Fig. 1.

production and functioning of the cell, the genes present in these pathways were expected to be highly expressed. In heterotrophic strains, genes affiliated with oxidative phosphorylation were also highly expressed. In contrast, photosynthetic pathways are particularly strongly expressed in the phototrophic strain *Synura* (strain LO234KE) and in the mixotrophic strain *Uroglena sp.* (strain WA34KE). Furthermore, energy metabolism and amino acid metabolism were particularly highly expressed in *Apoikiospumella mondseeiensis* (JBM08) and *Uroglena* strain WA34KE.

Principal component analysis (PCA) based on normalized gene expression values revealed that phylogenetically closely related strains presumably belonging to the same species, such as the *Poteriospumella lacustris* strains JBM10, JBNZ41 and JBC07, tend to cluster (Fig. 5). In contrast, different species—even though closely related—are scattered across the plot. For instance, the mentioned *Poteriospumella lacustris* strains as well as *Poterioochromonas malhamensis* strain DS, *Cornospumella fuschlensis* strain AR4D6 and *Acrispumella msimbaziensis* strain JBAF33 all belong to the C3 cluster in molecular phylogenies (Fig. S1; *Grossmann et al. (2016)*); yet they are dispersed across the plot. Mixotrophic and heterotrophic strains were separated in the PCA. This separation by trophic mode is largely visible along the second principal component which accounts for 10.36% of the total variation. The separation is consistent even within trophic modes: mixotrophic strains which are largely relying on heterotrophic nutrition such as *Poterioochromonas malhamensis* strain DS cluster close to the heterotrophic strains. Conversely, the heterotrophic *Cornospumella fuschlensis* (strain A-R4-D6), which (based on electron microscopical evidence) possesses a largely preserved plastid (L Grossmann, pers. comm., 2016), clusters close to the mixotrophic strains. In this latter strain major plastid-targeting genes are present and transcribed including almost complete gene sets

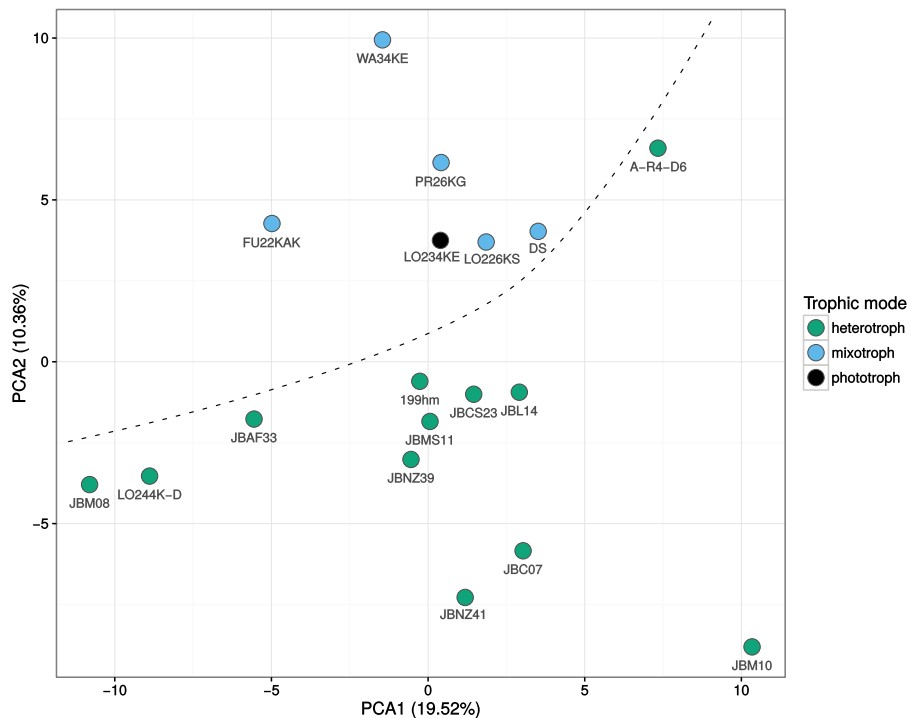

**Figure 5 Principal component analysis (PCA) of normalized expression profiles.** Depicted are the first and second component, which separate the mixotrophic (blue) and heterotrophic (green) strains, indicated by the dashed line, while the phototrophic (black) lies on the border of the mixotrophic group. The first and second component together explain 29.88% of the variance.

of the operational modules for the photosystem I and II. A similar clustering based on nutritional modes and phylogenetic relationship was observed previously for a larger set of 41 protistan genomes and transcriptomes by *Koid et al. (2014)*. Due to different growth requirements of the different taxa, a number of different growth media was used for the cultivation of strains. In order to exclude effects of medium composition and of food bacteria, we performed a likelihood ratio test comparing a full model including nutritional strategy, axenicity and medium against a reduced model including only the nutritional strategy. The additional factors of the full model did not have a significant effect on overall gene expression (observed adjusted $p$-value > 0.1), suggesting no impairing influence. However, we cannot fully exclude that the medium might nonetheless be a confounding variable. Despite the unknown influence shown in the first principal component, which at least partially can be attributed to phylogenetic relationship of the strains, the clear separation of mixotrophic and heterotrophic taxa points to systematic differences between the trophic modes. The separation is on the one hand due to differences in gene expression, but also depends on the group-specific presence or absence of genes in either heterotrophic or mixotrophic taxa. We will focus on the two aspects (1) group-specific genes and (2) differences in gene expression of common genes in the following paragraph.

## Influence of nutritional strategies

Heterotrophic chrysophytes presumably evolved from photosynthetic ancestors. This results in the conclusion that heterotrophy is on the one hand a reduction at the cellular level, the reduction of the plastid, and along with that a reduction of metabolic pathways associated with the plastid, i.e. photosynthesis, chlorophyll and carotenoid biosynthesis. On the other hand, a heterotrophic mode of nutrition requires new sources of metabolites (carbon, nitrate and phosphate) and therefore mechanisms for the uptake of essential nutrients by ingestion of prey organisms (*Boenigk & Arndt, 2000*; *Zhang et al., 2014*). Consequently, different nutritional strategies require distinct molecular pathway compositions. Especially pathways associated with the carbohydrate, energy and amino acid metabolism as well as vitamin biosynthesis (*Liu et al., 2016*) are affected by diverging nutritional strategies. We expect to observe such changes at the gene content level, where genes were lost in heterotrophic organisms, as well as at the expression level for genes participating in energy and biosynthesis pathways.

### Gene content analysis

The total number of orthologous genes (KOs) to which at least one transcript could be mapped (over all samples) is 3,635. Of these, 180 KOs only appear in mixotrophs, 89 only in phototrophs, 758 genes only in heterotrophs and 1,411 core genes are present in all three groups (Fig. 6). Since the phototrophic strains are only represented by one sample, presumably more phototroph-specific genes exist that are missing in the study due to low coverage. In general, the presence of genes only in one of the groups can be due to missing corresponding genes in the other groups, to no expression of these genes in the current state of the cell, or to a very low expression level that was not detectable by sequencing.

*General findings.* Gene contents systematically differed between heterotrophic, mixotrophic and phototrophic strains (Table 2). In particular, phototsynthesis-related pathways were enriched in phototrophic and mixotrophic strains, whereas pathways acting in nutrient absorption, biosynthesis and environmental sensing were enriched in heterotrophic strains.

*Photosynthesis.* Photosynthesis as well as glycine, serine and threonine metabolism and protein export pathways were enriched for the phototrophic strains. The pathway "photosynthesis—antenna proteins" was enriched in mixotrophic strains. At the level of individual strains we found various degrees of reduction in pathways associated with photosynthesis. All phototrophic strains, including the mixotrophs, expressed genes for the light harvesting complex. For the phototrophic strain and the mixotrophic strains we identified almost all genes of the photosynthesis pathway with minor reduction of genes in the photosystem I for the mixotrophs. Further, Ribulose-1,5-bisphosphate carboxylase/oxygenase (RuBisCo) was present in the transcriptomes of the mixotrophic and phototrophic strains. In contrast, none of the heterotrophic strains expressed genes involved in photosynthetic carbon fixation. However, photosynthetic pathways were in different stages of reduction among the heterotrophic strains: The two heterotrophic strains *Cornospumella fuschlensis* A-R4-D6 and *Pedospumella sinomuralis* JBCS23 are the only two
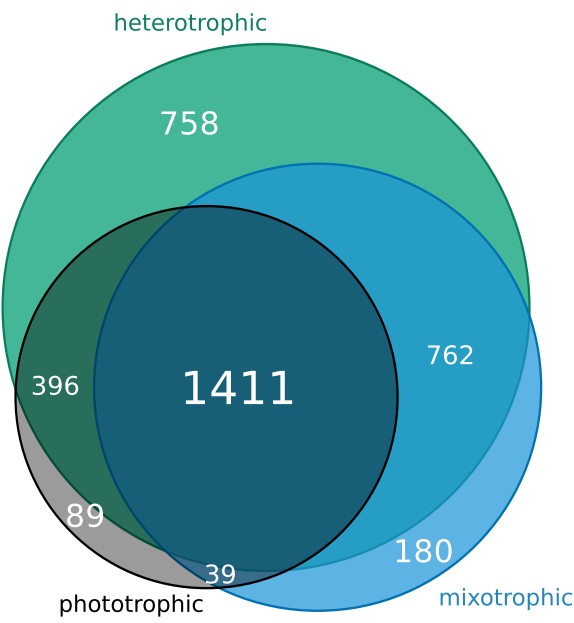

**Figure 6  Gene content analysis.** In our group of samples, the trophic groups share a core genome of 1,411 orthologous genes; phototrophs have in addition 89, mixotrophs 180 and heterotrophs 758 group-specific genes.

heterotrophic strains which express genes involved in the light harvesting complex (Lhca1, Lhca4). These two strains also expressed few genes of the photosystem I and II, cytochrome b6/f complex, electron transport and F-type ATPase. These findings indicate that the loss of photosynthesis in these two strains was relatively recent. For the photosynthetic protist *Euglena gracilis* it could be shown, that components of the photosynthetic bf complex have migrated from the chloroplast to the nucleus (*Torres et al., 2003*). Possibly this might also have occurred in the ancestors of A-R4-D6 and JBCS23 before the loss of pigmentation. The expression of various genes of photosynthetic pathways correspond with electron microscopical evidence for a relatively large plastid in *Cornospumella fuschlensis* A-R4-D6. The above two strains as well as *Poteriospumella lacustris* strains JBC07 and JBNZ41 express still one gene involved in the electron transport (PetH). These four strains thus have some remnants that hint to a functioning cyclic electron transport whereas genes affiliated with the photosynthesis pathways were not expressed in the other heterotrophic strains except for the general enzyme F-type ATPase which is, however, presumably not specific for photosynthesis pathways.

Another reduction can be seen in protein export pathways. The phototrophic strain expressed five exclusive genes for the SRP (signal recognition particle), which could be an indication of protein transport through the chloroplast membrane for pathways taking place in the plastid.

Taken together, the reduction of photosynthesis seems to start with the reduction of cost-intensive enzymes and pathways such as carbon fixation by RuBisCo which seems to be abandoned first and is missing in all investigated heterotrophs. The next step seems to be the reduction of photosystem I and II whereas genes for the photosynthetic

**Table 2** **Significantly enriched pathways (*p*-value < 0.001) in the sets of trophic-group-specific genes.** Enriched pathways for genes that are only present in one of the trophic modes are grouped according to KEGG hierarchy.

| Hierarchy I | Hierarchy II | Pathways |
| --- | --- | --- |
| **Pathways enriched in genes specific to heterotrophic organisms** | | |
| Cellular Processes | Cell motility | Regulation of actin cytoskeleton |
| Environmental Information Processing | Membrane transport | ABC transporters, Bacterial secretion system |
| | Signal transduction | Two-component system, MAPK signaling pathway |
| Metabolism | Amino acid metabolism | Arginine and proline metabolism, Lysine degradation, Tryptophan metabolism, Histidine metabolism |
| | Carbohydrate metabolism | Amino sugar and nucleotide sugar metabolism, Starch and sucrose metabolism, Pyruvate metabolism, Fructose and mannose metabolism |
| | Lipid metabolism | Glycerolipid metabolism |
| | Metabolism of cofactors and vitamins | Ubiquinone and other terpenoid-quinone biosynthesis, Folate biosynthesis, Nicotinate and nicotinamide metabolism |
| | Metabolism of terpenoids and polyketides | Geraniol degradation |
| | Nucleotide metabolism | Purine metabolism |
| | Xenobiotics biodegradation and metabolism | Bisphenol degradation |
| Organismal Systems | Digestive system | Bile secretion, Fat digestion and absorption |
| | Nervous system | Neurotrophin signaling pathway |
| | Sensory system | Phototransduction |
| **Pathways enriched in genes specific to mixotrophic organisms** | | |
| Cellular Processes | Cell communication | Focal adhesion |
| | Cell growth and death | Cell cycle - yeast |
| Environmental Information Processing | Signal transduction | TNF signaling pathway |
| Metabolism | Energy metabolism | Photosynthesis - antenna proteins |
| Organismal Systems | Immune system | NOD-like receptor signaling pathway |
| **Pathways enriched in genes specific to phototrophic organisms** | | |
| Genetic Information Processing | Folding, sorting and degradation | Protein export |
| Metabolism | Amino acid metabolism | Glycine, serine and threonine metabolism |
| | Energy metabolism | Photosynthesis |

electron transport are still present in a number of heterotrophs and seem to be reduced in a later step.

*Nutrient absorption and biosynthesis.* For heterotrophic strains we see pathways that hint to an absorption of nutrients from the feeding on bacteria and bacteria size organism as well as uptake of dissolved organic matter as carbon resource namely: fat digestion and absorption, ABC transporters, bile secretion, two component system and also possible homologous genes or acquisition of the bacterial secretion system pathway by horizontal gene transfer. Particularly, various transport functions are only present in heterotrophs including transporters for minerals and organic substances such as nitrate/nitrite, monosaccharides, phosphate and amino acid, peptides, metal etc. Complexes from the two-component

system that respond to phosphate limitation, regulate nitrogen assimilation, short chain fatty acid metabolism and amino acid uptake further indicate an increased nutrient and metabolite uptake. The presence of membrane fusion proteins and homologs to bile enzymes, responsible for the digestion, transport and absorption of fats, vitamins, organic compounds and the elimination of toxic compounds such as microcystins in heterotrophic chrysophytes are necessary adaptations to obtain carbon and energy from the ingestion of small organisms. Carbohydrate metabolism in general is enriched in group-specific genes for the heterotrophs. In a comparative transcriptome analysis performed on prymnesiophytes and stramenopiles *Koid et al. (2014)* likewise identified differences in carbohydrate transport and metabolism between hetero-, mixo- and phototrophic species and attributed the differences to a great diversity of isoenzymes to process and digest different sugars synthesized by prey. Apart from these, we found further differences in metabolic pathways, where whole subpathways were present only in one of the groups. Again, we found links to electron-transfer via the ubiquinone and other terpenoid-quinone biosynthesis pathway, where menaquinone is an obligatory component of the electron-transfer pathway and only present in heterotrophs. An increased production of glutamin and glutamate is observed in histidine metabolism, arginine and proline metabolism. Glutamin and glutamate can function as substrate for protein synthesis, precursor for nucleotide and nucleic acid synthesis and precursor for glutathione production (*Newsholme et al., 2003*) and indicate a maximal growth rate in heterotrophs (*Boenigk, Pfandl & Hansen, 2006*). Further differences in amino acid production include the production of cysteine and methionine. While sulfate assimilation is essential for phototrophic growth to produce cysteine and methionine, it is usually absent in organisms that ingest sulfur containing cysteine and methionine (*Kopriva et al., 2008*). Therefore, heterotrophic chrysophytes should be able to obtain reduced sulfur compounds from ingested prey. Still, we find the energy consuming assimilatory pathway in all trophic groups. In the phototrophic organism it is used to generate methionine from cysteine, via the existent homocysteine S-methyltransferase, but enyzmes for the synthesis of cysteine from methionine are absent as is known for plants. In the mixotrophic and heterotrophic organisms enzymes for the production of both amino acids are present. The reaction from homo-cysteine to methionine is possible using cobalamin-dependent or -independent methyltransferases, metH and MetE respectively. Both of these are found for all trophic modes, but only the heterotrophic chrysophytes possess several genes of the cobalamin synthesis pathway. These could have been acquired from bacteria through ingestion or from a form of symbiosis as identified in *Chlamydomonas nivalis* (*Kazamia et al., 2012*).

*Environmental sensing.* Heterotrophic strains possess numerous genes from the two-component system, an environmental-sensing two-component phosphorelay system that has been identified in archae, bacteria, protists, fungi and plants (*Simon, Crane & Crane, 2010*). These include systems from the chemotaxis family for surface or cell contact, triggering extracellular polysaccharide production, for twitching motility and flagellar rotation due to sensing of attractants or repellents. Cells interact with their environment in various ways. They secrete a great variety of molecules to modify their environment, to

protect themselves or to interact with other cells. Genes for lipopolysaccharide biosynthesis produce glycoproteins, possibly as a coating layer to better protect the heterotrophic strains that are more resistant to environmental stresses. Additionally, we exclusively find the lysine decarboxylase in the heterotrophic strains, which catalyses the reaction of L-lysine to cadaverine. Cadaverine is an intermediary product in the synthesis of alkaloids, that could serve as protection against feeding. These are possibly secreted by enriched genes of the general secretion pathway. Additional signal transduction pathways and pathways from the sensory system (phototransduction) are possibly also engaged in environmental information processing. MAPK family members have been identified in lower eukaryotes such as *Chlamydomonas reinhardtii* and are known to be important signaling molecules that perceive various signals and transduce them for active responses to changing environmental conditions (*Mohanta et al., 2015*). We conclude from the presence of orthologous genes from these pathways that similar functions might be performed in heterotrophic chrysophytes and explain the higher motility of heterotrophic chrysophytes that have to sense and find bacterial food.

### Gene expression analysis

Apart from the presence/absence information of genes (gene content), we analysed changes in the relative abundance of KOs on the intersection of KOs present in mixo- and heterotrophic strains. The single phototrophic strain was excluded from this analysis. We found 67 out of 2,134 KOs to be significantly differentially expressed ($p < 0.1$ after Benjamini–Hochberg correction with DESeq2). These KOs are listed in Table S1 with their gene symbol, gene name, log fold-change, $p$-value and adjusted $p$-value. A pathway enrichment analysis was performed for the significant genes to detect overrepresented associations with specific pathways (see Methods). Visual inspection of all pathways coloured according to differential expression was performed. The pathways with significant differences ($p$-value $< 0.1$) are shown in Fig. 7 with color-coded KOs that are significantly differential between the mixotrophic and heterotrophic group.

*General findings.* Genes involved in energy metabolism, particularly pathways dealing with photosynthesis are significantly differentially expressed in mixotrophic strains, such as carotenoid biosynthesis, photosynthesis and porphyrin and chlorophyll metabolism. In heterotrophic strains pathways with higher expression in energy metabolism include the oxidative phosphorylation. In addition, we find enriched pathways and differentially expressed genes acting in steroid biosynthesis and the amino acid metabolism such as glutathione metabolism (Fig. 7 and Table S1).

*Energy metabolism.* Most differences in photosynthesis pathways between mixo- and heterotrophs were already discussed in the gene content analysis. Additionally, we identified differentially expressed genes in related pathways including the porphyrine and chlorophyll metabolism, carotinoide metabolism and retinole metabolism. Here, genes are still present in some heterotrophs but show a reduced expression such as magnesium-protoporphyrin O-methyltransferase and protochlorophyllide reductase in the porphyrin and chlorophyll metabolism or PsbE, PsbO, PsbQ, PsbV and PsbB in the photosystem I and II. We

**Figure 7** **Pathways enriched for differentially expressed KOs between hetreotrophic and mixotrophic strains.** Each row represents a pathway; the cells in a row represent KOs of the respective pathway. KOs showing a significant difference in expression (Benjamini–Hochberg adjusted $p$-value $< 0.1$) between trophic modes were used in a pathway enrichment analysis. For enriched pathways ($p$-value $< 0.1$), all KO IDs belonging to the pathway are shown, and significantly differential KOs are colored with their log-fold change. Yellow indicates a higher expression in the mixotrophs while blue shows a higher expression of the gene in heterotrophs.

further see an enrichment of higher expressed genes in mixotrophs for the lower part of glycolysis. At this point a product of photosynthsis, glycerate-3-phospate enters the glycolysis and TCA cycle as carbon source. These pathways are likely used to generate energy for cell maintenance, since biosynthesis processes related to growth are not upregulated in mixotrophs. In contrast, heterotrophs show higher expression of genes involved in oxydative phosporylation, e.g., cytochrome c oxidase COX10, NADH dehydrogenase NDUFS5 and F-type ATPase, which could indicate higher respiration rates in heterotrophs. It is well known that there are variations in respiration rates among different protist species, physiological conditions and cell sizes. Under comparable conditions smaller cells have a higher rate of living and an increased metabolic rate (*Fenchel, 2005*) which is closely coupled to growth and reproduction. This is complying with the reduced cell size in heterotrophic strains of around 5 µm (*Grossmann et al., 2016*) and higher growth rates for heterotrophic species under suitable food conditions (*Boenigk, Pfandl & Hansen, 2006*). Additionally, the remnants of an operational photosynthetic cyclic electron transport in some heterotrophs without functional plastids suggest a functional adaptation similar to certain bacteria that adapted cytochrome chains, previously used to produce ATP in photosynthesis, for the generation of ATP by oxidative phosphorylation (*Sleigh, 1989*).

*Amino acid metabolism.* In addition to the observation in the cysteine and methionine pathway described before, parts of the methionine salvage pathway are higher expressed for heterotrophs. The methionine salvage cycle is used to recycle sulfur, which otherwise has to be obtained using the energy consuming assimilatory pathway. Another source or sulfur is glutathione, which is a major reservoir of non-protein reduced sulfur and enriched

in heterotrophs (*Mendoza-Cózatl et al., 2005*). Further, in the glutathione metabolism the ornithine decarboxylase shows higher expression possibly leading to an increased production of the polyamines putrescine and spermidine. Their concentration is increased during growth and high metabolic activity and elevates the rates of DNA, RNA and protein synthesis (*Ahmed, 1987*) indicating growth.

In addition, we found differences in expression for the glutamate metabolism. For example, the expression of glutamate synthase (gene K00284) is strongly decreased in heterotrophs (Table S1). For *E. coli* it was shown that it utilizes two ways to form glutamate. These differ in the fact that one way (glutamate dehydrogenase) is energetically efficient (no direct requirement for ATP), while the other one (GOGAT pathway: glutamine synthetase plus glutamate synthase) uses ATP (*Helling, 2002*). The choice among these parallel pathways in biosynthesis has been hypothesized to control the speed and efficiency of growth. Similarly in the oxidative phosphorylation pathway, enriched for significant differentially expressed genes, the choices of dehydrogenase and oxidase control the efficiency of ATP synthesis (*Helling, 2002*). Differences in expression and gene content in the above pathways point to known divergences in growth rates for mixo- and heterotrophic species (*Boenigk, Pfandl & Hansen, 2006*).

Related to the amino acid metabolism we find higher expressed lysosomal enzymes for the break down and usage of incorporated biomolecules for the heterotrophic group.

*Lipid metabolism.* The ability of vitamin D production was shown to vary between different algal species (*Jäpelt & Jakobsen, 2013*). In chrysophytes, ergosterol (the provitamin form of vitamin $D_2$) was found in an early study by *Halevy, Avivi & Katan (1966)* in *Ochromonas danica*. For heterotrophic chrysophytes we see in the steroid biosynthesis pathway an increased transcription of genes responsible for ergosterol production, but from the transcriptome it remains unknown whether ergosterol is used to protect the cell against uv radiation by the conversion into vitamin $D_2$.

## Phylogenetic inference from assembled transcripts
### Overview
Despite their functional similarity, heterotrophic chrysophytes have evolved several times independently from phototrophic organisms by a reduction of their plastid genome. The most recent phylogeny based on the SSU rDNA (*Grossmann et al., 2016*) shows this for some of the presented strains.

We aim at developing a method to use available transcriptome data to place strains into their evolutionary context and reconstructing their phylogenetic relationship. Especially when marker genes are not yet sequenced, but transcriptome data is readily available, this information can be valuable.

Unfortunately, short-read transcriptomic data pose several problems for phylogenetic inference which render alignment-based multigene approaches diffcult and inefficient. These problems include the presence of several alternative transcripts per gene, difficulties in the identification of orthologous genes from assembled contigs in all strains, or the correct detection of overlapping regions in the contigs for multiple sequence alignments.

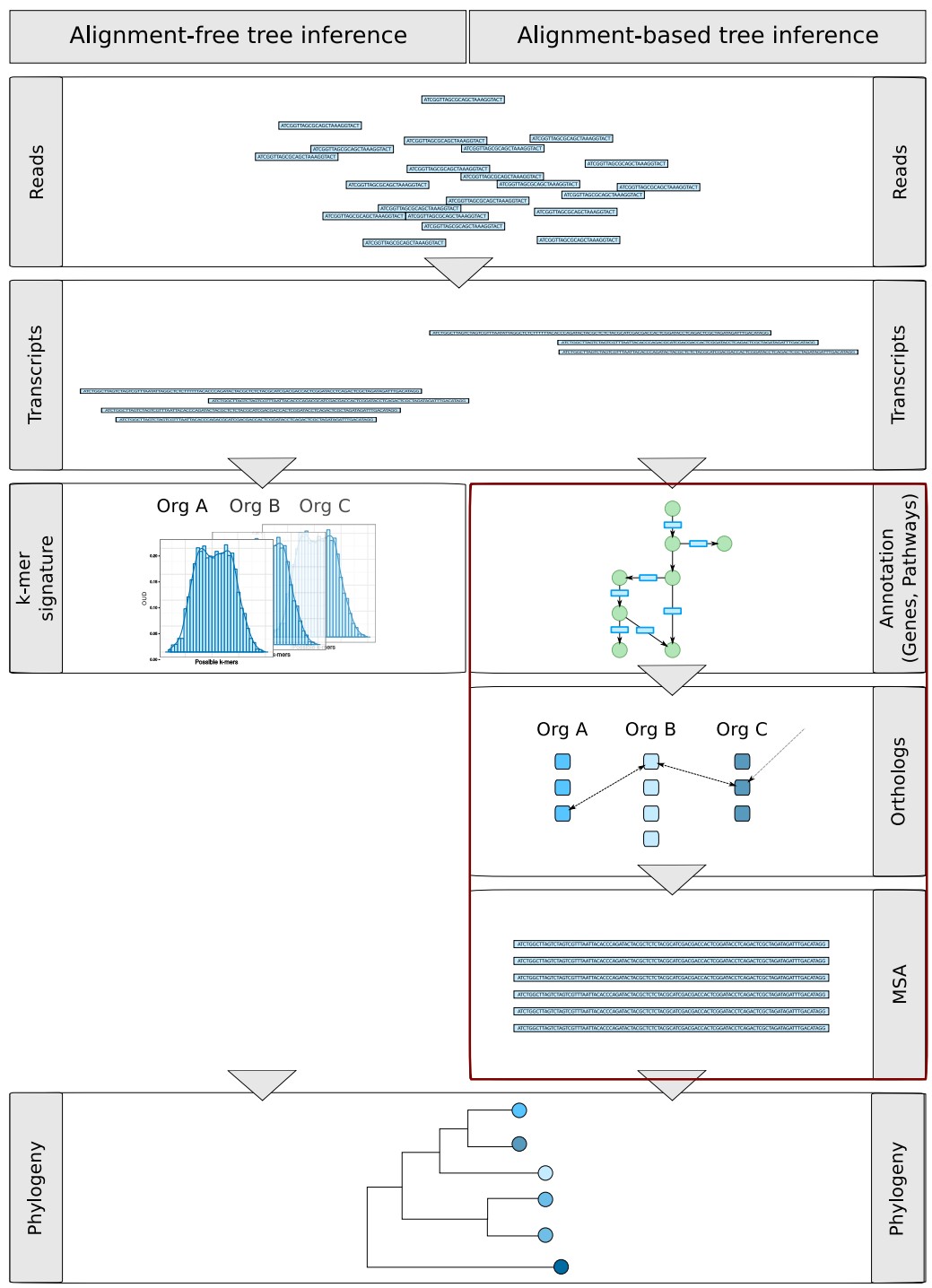

**Figure 8** **Overview of alignment-free and alignment-based gene-centric approaches of phylogenetic inference.** Both approaches include the assembly of reads to transcripts using Trinity (*Grabherr et al., 2011*). Based on the transcript sequences in the alignment-free approach, *k*-mers were counted (4-mers and 6-mers) and used to calculate transcriptome signatures. Using the euclidean distance between the signatures, trees were constructed using the Unweighted Paired 

**Figure 8 (…continued)**
Group Method with Arithmetic Mean (UPGMA). In contrast, for the alignment-based approach, the steps marked in red differ. Here the transcripts were annotated with KEGG genes, pathways and KEGG orthology information (*Kanehisa & Goto, 2000*). The KEGG orthology information was used to find orthologous genes between all strains. Multiple sequence alignments were constructed with MAFFT (*Katoh & Standley, 2013*) using overlapping regions of the transcripts of all 18 strains. One or several transcripts, constituting splice variants or alternative assemblies, were included if they feature an adequate length. Based on the multiple sequence alignments, a model test was performed and a bootstrapped maximum-likelihood phylogeny estimated.

As a result, one may obtain few and short multiple sequence alignments, i.e., a weak basis for alignment-based phylogenetic inference. Indeed, we identified only few genes with known function that were present in all 18 strains, since some (e.g., FU22KAK and JMB08) possessed only few and shorter transcripts due to quality issues (see Assembly of transcripts). Even fewer genes generated long enough multiple sequence alignments, without alternative exons, for the calculation of phylogenetic trees. In addition, alignment-based methods are comparatively slow.

We therefore adapted a fast alignment-free $k$-mer based approach for *genomes* (*Reva & Tümmler, 2004*; *Pride et al., 2006*; *Patil & McHardy, 2013*; *Chan & Ragan, 2013*; *Fan et al., 2015*) to work on assembled *transcriptomes*. Note that up to now, it is not established that alignment-free approaches intended for genome-wide use produce reasonable results on assembled transcriptomes. Therefore, we here present a comparison between phylogenetic trees based on the SSU gene, on a multiple alignment of selected high-quality assembled gene sequences and on k-mer methods on all assembled transcripts.

### Alignment-based and Alignment-free phylogenetic inference from transcriptomes

Figure 8 outlines the steps of the alignment-free and alignment-based approaches. Both approaches include the assembly of reads to transcripts using Trinity (*Grabherr et al., 2011*).

For the alignment-based approach, transcripts were annotated with KEGG genes, pathways and KEGG orthology information (*Kanehisa & Goto, 2000*). The KEGG orthology information was used to find orthologous genes between all strains as a faster alternative to pair-wise bi-directional BLAST searches. Overlapping regions between all 18 strains are detected, which are thereupon used to construct multiple sequence alignments with MAFFT (*Katoh & Standley, 2013*). One or several transcripts, constituting splice variants or alternative assemblies, were included if they feature an adequate length. Based on the multiple sequence alignments at the nucleotide level, a model test was performed and a bootstrapped maximum-likelihood phylogeny estimated. In general, the complete workflow can take hours to days, depending on the number of species and number of orthologous genes.

In contrast, the alignment-free approach does not need orthology information or multiple sequence alignments, omitting the steps marked in red in Fig. 8. Based on the transcript sequences, $k$-mers were counted (4-mers and 6-mers) and used to calculate transcriptome signatures (Eq. (1)). Using the Euclidean distance between the signatures,

trees were thereupon constructed applying the Unweighted Paired Group Method with Arithmetic Mean (UPGMA).

In the following, we describe how we computed transcriptomic signatures. We first computed normalized oligonucleotide usage deviations (OUDs), the ratio of observed excess counts of $k$-mers in a transcriptome to the expected count under a null model, following *Reva & Tümmler (2004)*.

To determine the oligonucleotide usage deviations (*OUD*) among transcriptomes, the observed number $N(z)$ of each $k$-mer $z$ is compared to its expected value and normalized. We define

$$OUD(z) := \frac{N(z) - E_1(z)}{E_0(z)}, \tag{1}$$

where $E_0(z) = (L-k)/4^k$ is the expected count of $k$-mer $z$ assuming uniform distribution of all $k$-mers in a transcriptome of length $L$, and $E_1(z)$ is the expected count of $k$-mer $z$ using mononucleotide content, corresponding to an i.i.d. model (independent identically distributed).

We computed tetranucleotide and hexanucleotide signatures containing $4^4 = 256$ and $4^6 = 4{,}096$ elements using Jellyfish (v1.1.2; *Marçais & Kingsford (2011)*). The Euclidean distance between two signatures $x$ and $y$ of $4^k$ possible $k$-mers was used, defined as

$$d(x,y) = \sqrt{\sum_{k\text{-mers } z} (OUD_x(z) - OUD_y(z))^2}. \tag{2}$$

The abundance of tetra- and hexanucleotides was calculated over the transcripts that were de-novo assembled. This was done on (A) all transcripts, (B) the longest ORF of the coding regions within transcripts obtained with TransDecoder (*Haas, 2013*) to prevent multiple counts for several transcripts of one gene and (C) the longest ORFs of genes present in all strains to remove genes that were present due to nutritional strategies, but developed and got lost independently several times during evolution.

The Unweighted Paired Group Method with Arithmetic Mean (UPGMA, phangorn package by *Schliep (2011)*) was used with the pairwise Euclidean distances between all chrysophyte transcriptome signatures to construct the phylogenetic tree. Bootstrapped phylogenies were constructed by bootstrapping the OUDs before distance and tree calculation and counting the number of bipartitions identical to the original phylogenetic tree (ape package by *Paradis, Claude & Strimmer (2004)*).

### Comparison of trees

The application of 4-mer or 6-mer signatures resulted in the same phylogeny, which is shown in Figs. 9A–9C. The $k$-mer phylogenies were calculated on all transcripts (A), on predicted coding sequences (CDS) in the longest open reading frame (ORF) of each gene (B) and on coding sequences of genes that are present in all samples (C). Bootstrap values are shown for the inner nodes of the trees.

We used the longest transcript of each orthologous gene to calculate multiple sequence alignments for the alignment-based approach. The multiple sequence alignments were

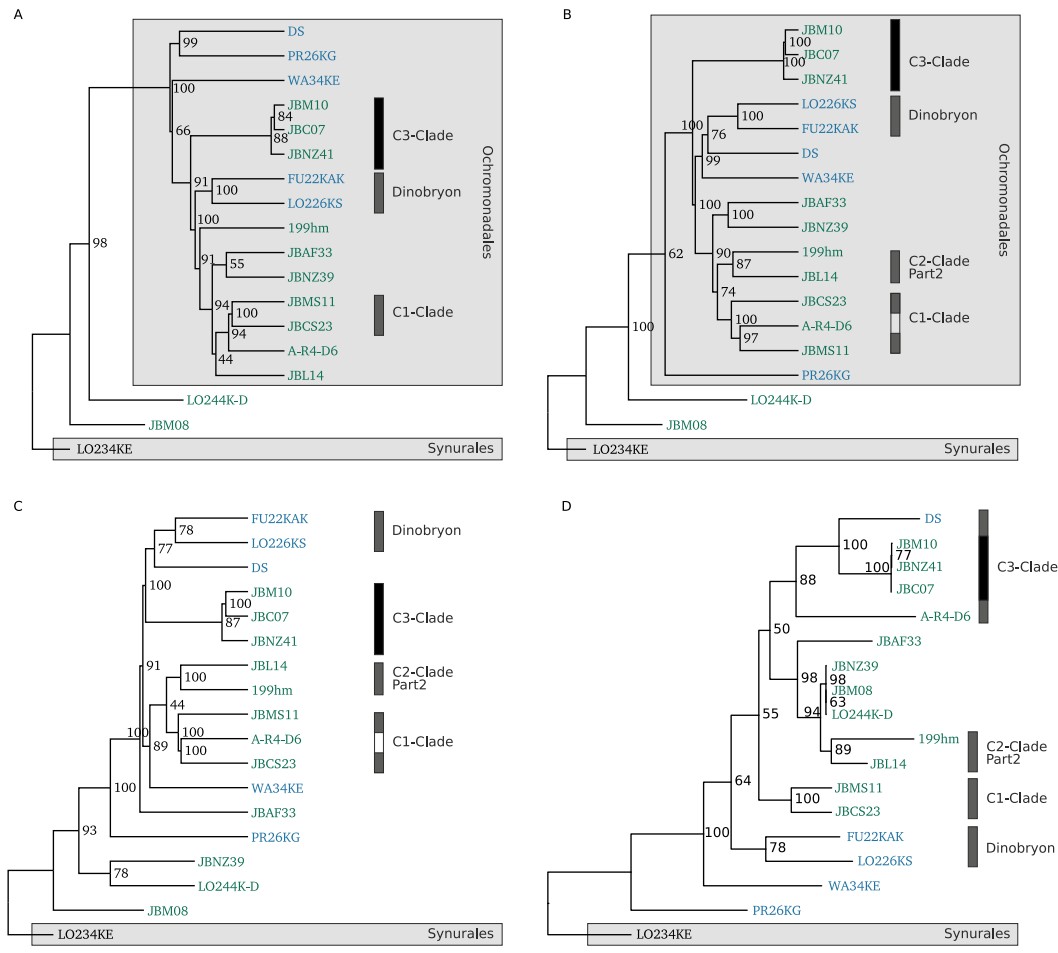

**Figure 9  Inferred phylogenetic trees.** *K*-mer based approaches (A–C) and a multigene based approach (D) were used to calculate phylogenetic trees from transcript sequences. (A) shows the *k*-mer based phylogenetic tree calculated on transcript sequences, (B) on calculated CDS in transcripts and panel (C) on calculated CDS of transcripts that are present in all strains. (D) depicts the maximum likelihood phylogenetic tree from 8 concatenated multiple sequence alignments of transcript sequences. Bootstrap values are shown for the inner nodes of the trees in A–D (values > 50 are shown). Grey boxes indicate the genera, order and clades of the Chrysophyceae species. Strains are coloured according to their trophic mode.

generated by aligning transcripts to KEGG genes and pruning the alignments to regions of overlapping sequences between all 18 strains. These were manually corrected and several genes were concatenated to create a 1,968 bp-long multigene multiple sequence alignment for phylogenetic inference. In total, regions from eight genes were used including three genes with unknown functions, calmodulin, ADP-ribosylation factor 1 and three genes coding for ribosomal proteins. The resulting phylogeny is shown in Fig. 9D. In previous approaches we tried to use all sequences mapping to identical KEGG Orthologs. But the sequence divergence was too high and the transcript contigs covered different parts of the gene which prevented clustering and the construction of multiple sequence alignments. We therefore had to settle for fewer genes and this also led to the consideration of alignment-free approaches.

In contrast to these transcriptome phylogenies, the SSU phylogeny of considered strains is depicted in Fig. S1. We extended the known SSU rDNA phylogenetic tree (*Grossmann et al., 2016*) for five additional strains, including now 17 out of the 18 sequenced strains to use these as the gold standard phylogeny. For the last strain the SSU sequences could not be sequenced yet. The phylogenetic tree was calculated on a multiple sequence alignment of 1869 nucleotides (for details see Methods).

We rooted the transcriptome trees according to the SSU tree with *Synura sp.* (LO234KE) as outgroup, for which it is known from 18S analyses that it is very distantly related to the other species (*Grossmann et al., 2016*). In all transcriptome phylogenies we found the C3 clade of *Poteriospumella lacustris* strain JBC07, *Poteriospumella lacustris* strain JBNZ41 and *Poteriospumella lacustris* strain JBM10, which are known to be very closely related and probably represent the same species, as well as the *Dinobryon sp.* strain LO226KS and strain FU22KAK. The other members of clade 3 *Cornospumella fuschlensis* (A-R4-D6) and *Poterioochromonas malhamensis* DS only show as one clade in Fig. 9D, while *Acrispumella msimbaziensis* (JBAF33) clusters outside of the clade. Other closely related species such as *Pedospumella encystans* (JBMS11) and *Pedospumella sinomuralis* (JBCS23) (part of C1 clade) and *Spumella vulgaris* (199hm) and *Spumella bureschii* (JBL14) cluster together in B, C and D. The separation between Ochromonadales, Hydrurales and Synurales is present in A and B, while for fewer genes (C) and the multigene based approach (D) it is perturbed. Considering that the *k*-mer tree for all transcripts (A) probably overestimates *k*-mers that are present in a gene with many transcripts, the *k*-mer tree on the longest ORF per gene (B) should better reflect the true phylogeny, showing in the higher resemblance to the SSU phylogeny with more identical clades and separation of higher orders. Using only CDS of genes that are present in all samples (221 genes) for the calculation of oligonucleotide usage deviation did not improve the phylogeny. Noticeably, two strains are always displaced in the *k*-mer phylogenies which are *Poterioochromonas malhamensis* (DS) and *Cornospumella fuschlensis* (A-R4-D6). Their phylogenetic positioning is either superimposed by their nutritional mode or by their GC content, since DS always clusters with PR26KG and/or WA34KE which have a very similar GC content. However, a normalization using dinucleotide content did not improve the *k*-mer phylogeny.

For the evolutionary-close groups the multigene tree is highly similar to the SSU tree, while it clusters some of the strains, such as *Spumella lacusvadosi* (JBNZ39), *Apoikiospumella mondseeiensis* (JBM08), *Ochromonas* or *Spumella sp.* LO244K-D with a high similarity. These strains are not closely related in any other phylogeny and *Apoikiospumella mondseeiensis* is not part of Ochromonadales. The strain LO244K-D was isolated and morphologically classified as an *Ochromonas* species (J Boenigk, pers. comm., 2016), where it also resides using the k-mer approaches. But according to new findings inferred from the SSU sequence, it is closely related to *Poteriospumella lacustris*.

The single gene phylogenies (Fig. S1) could not distinguish the strains of *Poteriospumella lacustris* probably due to a high conservation in the gene. All *k*-mer based phylogenies indicate a higher similarity between *Poteriospumella lacustris* strain JBM10 and *Poteriospumella lacustris* strain JBC07. The same results were found by *Stoeck, Jost & Boenigk (2008)* using the SSU and a concatenation of sequences for three protein-coding

genes (alpha-tubulin, beta-tubulin and actin) and rDNA fragments (SSU rDNA, ITS1, 5.8S rDNA, ITS2). In contrast, the multigene approach shows a higher similarity between *Poteriospumella lacustris* strains JBM10 and JBNZ41.

Overall, for closely related species in the same genus or a taxonomy of higher orders the *k*-mer approach on coding sequences worked best, but it has to be noted that it did not reproduce the SSU phylogeny nor did the other three phylogenetic approaches. Evaluation on the SSU rRNA-based methodology has shown that SSU phylogenies are sufficiently reliable and robust for evaluating relationships between organisms related at the genus level and higher (*Liu & Jansson, 2010*). Therefore, the current SSU phylogeny will probably also undergo changes with new sequences becoming available, but at the moment depicts the most reliable phylogeny at the genome level and higher. Furthermore, the Chrysophyceae exhibit a high phylogenetic diversity with up to 10% pairwise differences in nucleotides in the SSU sequence and it was shown previously that the SSU rRNA gene offers here a higher resolution than genome-based approaches (*Liu & Jansson, 2010*). Therefore, probably neither single gene alignment-based approaches nor transcriptome- or genome-based approaches can resolve nodes at all taxonomic levels and its efficacy will vary among clades. Depending on the intended resolution the methodology and chosen gene, considering sequence conservation, have to be adapted.

Interestingly, evolutionarily closely related species such *Pedospumella* strains JBMS11 and JBCS23; *Spumella bureschii* (JBL14) and *Spumella vulgaris* (199hm); *Poteriospumella lacustris* strains JBM10, JBC07 and JBNZ41 and outgroups in the phylogeny such as strains LO244K-D and JBM08 also cluster closely together in the functional analysis (see Fig. 5). While, the functional separation of heterotrophic and mixotrophic strains is mostly absent in the phylogenetic analysis, solidifying a single evolutionary origin of heterotrophic chrysophytes as unlikely.

We do not propose to sequence transcriptomes as a replacement for the creation of phylogenies from marker genes. Instead, we suggest to use the multitude of available transcriptome data as an addition to marker genes for the inference of phylogenetic relationships. To use assembled short-read mRNA sequences properly for phylogenetic inference requires a new methodology, without the necessity to create multiple sequence alignments. Here the proposed *k*-mer approach comes into effect, which additionally provides several benefits. One of them is its speed. The signatures and distances of the alignment-free method can be calculated within a few minutes (the CPU time of transcript 6-mer tree calculation with 100 bootstraps was 7 min 10 sec) and do not depend on orthology identification or construction and manual correction of multiple sequence alignments. This is the biggest issue in phylogenetic reconstruction using transcript sequences and difficult to do correctly due to uncertainties during the assembly phase and alternative transcripts. Further, a low sequencing depth of one sample will only influence its phylogenetic placement, but not the construction of the entire phylogeny. In contrast, in gene-based approaches long-enough MSAs could not be constructed in this case or only after removal of such samples. Lastly, when additional transcriptome sequences become available only the signatures and distances to the other oligonucleotide signatures have to be recalculated to include further taxa in the phylogeny, which is very efficient.

## CONCLUSIONS

Chrysophytes have for decades served as protist model species in ecology and ecophysiology since they play an important role as grazers and primary producers in oligotrophic freshwaters. Up to now few molecular analyses exist for chrysophytes, currently restricted to the reconstruction and analysis of phylogenies. Therefore, molecular data is also limited to marker gene sequences such as cytochrome oxidase subunit 1 (Cox1), 28S and 18S rRNA, ITS1 and few EST sequences. This study tries to extend this knowledge by sequencing whole transcriptomes of 18 chrysophyte strains. This data is used thereupon to characterize the strains, compare their physiology based upon varying degrees of loss of pigmentation and changes in nutritional strategies and analyse their phylogenetic relationship.

The essential pathways and processes are highly active in all strains, including ribosome maintenance, as well as pathways of the primary metabolism including oxidative phosphorylation, carbon metabolism, transcription and translation and for the photosynthetic strain—photosynthesis. Differences between organisms with different nutritional strategies are observed based on the presence and absence of genes and changes in gene expression. We find group-specific genes enriched in photosynthesis, photosynthesis—antenna proteins and porphyrin and chlorophyll metabolism for phototrophic and mixotrophic strains that can perform photosynthesis while genes involved in nutrient absorption, environmental information processing and various transporters (e.g., monosaccharide, peptide, lipid transporters) are present only in heterotrophic strains that have to sense, digest and absorb bacterial food. Additionally, for mixotrophic strains, we see a higher expression of genes participating in photosynthesis, such as carbon fixation in photosynthetic organisms, carotenoid biosynthesis, photosynthesis and porphyrin and chlorophyll metabolism. At the strain level, we observed for the photosynthesis pathways various degrees of reduction in essential complexes. For most heterotrophic organisms, the photosystem I and II are completely missing, whereas four of the heterotrophic strains still have some remnants that hint to a functioning cyclic electron transport, possibly transferred to the nuclear genome. In general, carbon fixation by ribulose-1,5-bisphosphate carboxylase/oxygenase seems to be abandoned first in all heterotrophs, followed by most genes necessary for the photosystem I and II. Genes for the photosynthetic electron transport are still present in some heterotrophs and seem to be reduced in a later step. In heterotrophic strains pathways with higher expression in energy metabolism including oxidative phosphorylation occur possibly due to higher respiration rates. We identified enriched pathways and differentially expressed genes acting in steroid biosynthesis—production of ergosterol—and the amino acid metabolism such as glutathione metabolism and cysteine and methionine pathway. Alternative reactions within the latter pathways, with varying energetic costs or gains, point to known divergences in growth rates for mixo- and heterotrophic species.

In addition to the comparison of chrysophyte physiology by trophic mode, we presented an alignment-free approach to use the transcriptomic sequences to infer phylogenetic relationships. We use a $k$-mer based approach which provides several benefits for transcripts assembled from short read RNA-Seq data. Our best result was obtained

using a mononucleotide-normalized 6-mer phylogenetic approach on coding sequences of the longest open reading frame per assembled component. The *k*-mer approach is consistent with SSU phylogenies in separating chrysophycean orders, i.e. Ochromonadales, Hydrurales and Synurales. Also similar to multigene phylogenies the *k*-mer approach does not resolve the precise branching order of these taxa. However, for intrageneric and intraspecific variation the *k*-mer strategy shows good results, resolving the phylogeny at the species level and below better than with using the SSU gene.

## ACKNOWLEDGEMENTS

We would like to thank Sarah Völker, Jenny Spangenberg and Meike Strybos for making the additional SSU sequences available, and Manfred Jensen for his discussions on metabolic pathways. We would also like to acknowledge the three anonymous reviewers whose comments and suggestions greatly improved this manuscript.

### Funding

DB, CB, BS, SR and JB were supported from the Deutsche Forschungsgemeinschaft (DFG) within the Priority Programme DynaTrait (SPP 1704), grants SU 217/14-1, RA 1898/1-1 and BO 3245/14-1. The funders had no role in study design, data collection and analysis, decision to publish, or preparation of the manuscript.

### Grant Disclosures

The following grant information was disclosed by the authors:
Deutsche Forschungsgemeinschaft (DFG) within the Priority Programme DynaTrait: SPP 1704.
SU: 217/14-1.
RA: 1898/1-1.
BO: 3245/14-1.

### Competing Interests

Sven Rahmann is an Academic Editor for PeerJ.

### Author Contributions

- Daniela Beisser analyzed the data, wrote the paper, prepared figures and/or tables, reviewed drafts of the paper, did statistical and bioinformatical analysis of sequencing data and interpreted results.
- Nadine Graupner analyzed the data, reviewed drafts of the paper, contributed to the analysis of metabolic pathways.
- Christina Bock performed the experiments, reviewed drafts of the paper, provided expertise on chrysophyte taxonomy.
- Sabina Wodniok performed the experiments, reviewed drafts of the paper.
- Lars Grossmann reviewed drafts of the paper, provided expertise on chrysophyte taxonomy.

- Matthijs Vos and Bernd Sures reviewed drafts of the paper, helped with interpretation of data.
- Sven Rahmann wrote the paper, reviewed drafts of the paper, provided bioinformatics expertise.
- Jens Boenigk conceived and designed the experiments, wrote the paper, reviewed drafts of the paper, led and guided the study.

### DNA Deposition

The following information was supplied regarding the deposition of DNA sequences:

The raw sequence data in FASTQ format and assembled transcripts are available at the European Nucleotide Archive (ENA) accession number PRJEB13662.

### Supplemental Information

Supplemental information for this article can be found online at http://dx.doi.org/10.7717/peerj.2832#supplemental-information.

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
