# Peer review of "Comprehensive transcriptome analysis provides new insights into nutritional strategies and phylogenetic relationships of chrysophytes"

_PeerJ, doi:10.7717/peerj.2832_

## Round 0.1 · original submission · Major Revisions

· Academic Editor

Major Revisions

As you can see, the reviewers have raised some very important issues for the analysis of your data set.

In order to facilitate their review of your revised manuscript, you must give a point-by-point response to these comments both in the text and in your reply.

Reviewer 1 ·

Basic reporting

The paper entitled “Comprehensive transcriptome analysis provides new insights into nutritional strategies and phylogenetic relationships of chrysophytes” by Beisser et al., is a motivated paper that provides new information, through transcriptome analysis, on the diverse nutritional strategies among protists. It is a good effort since the authors provide a strong and accurate bioinformatic and phylogenetic interpretation of the data, that lacks, however, a more robust explanation on the biology and possibly the biochemistry behind their findings. This somehow leads to my next concern which is the interpretation of their data and how nutritional strategies can be linked in an interesting way to evolution. Presenting the differences in terms of gene expression and gene content analysis is not sufficient if these findings and the presented pathways are not further explained in the evolutionary frame that the authors want to promote (at least as stated in the abstract and text). This can be achieved if these highlighted pathways, and especially the metabolic and energetic ones, are further analyzed in a more comparative way in the content of the biology and the physiology of the mixotropnic versus the heterotrophic and phototrophic chrysophytes.

As general comments I believe that they have a limited introduction and the material and methods section is really constrained with abbreviations which are not explained. I believe that these parts have to be more elusive. Also, the selection of the photoperiod as well as the light intensity have somehow to be justified either though references or previous data.
Another concern is the different media that the authors used. Even though they state that the data analysis - gene expression did not covary with media composition, however I am skeptical on this, since gene profiling is under the influence of different nutritional sources. The expression pattern drastically changes under different media since the metabolism of different components regulates the parts of chromatin that are accessible for transcription and thus the mRNA levels and the gene profiling. Based on this, I do not know how accurate is to compare gene data sets attributed to mixotrophy with those attributed to heterotrophy nutritional strategy, especially when some of these data sets derive from separate mixotrophic or heterotrophic strains that are cultivated in different media. At this point probably a statistical approach could be a good tool to show no statistical differences between the different growth media and the different mixotrophic or hetetrotrophic strains is observed, at least for the data sets that the authors present in the text.

However, as I already said, the paper of Beisser et al., gives novel information that can be further exploited. So I am positive to their work. I would request though to elucidate their introduction and the material section and to reinforce their discussion parts which are spread in the text.

Experimental design

I have made my comments on the previous section

Validity of the findings

Please see my comments on the basic reporting.

Additional comments

Good work with a lot of potential and new insights on the different nutritional strategies. It could be good if there was a link as I already said with the physiology (where information is applicable) and possibly with the ecology of the strains that you used. Try to see also for genes that are providing evidence for symbiosis between mixotrophs/heterotrophs and bacteria. Possible adaptations which may be presented in structures (e.g. different composition and type of lipids of membranes) in the mixotrophs or heterotrophs versus phototrophs that can be attributed to bacteria.

Reviewer 2 ·

Basic reporting

The manuscript is clearly written, the introduction cites relevant literature, the structure of the manuscript conforms to the discipline norm, and raw data are supplied.
The authors also state that assembled transcripts are provided. Although this is not always the case for this type of manuscript this information would be very useful; unfortunately I was not able to find these data with the provided accession number.
Figures are generally clear, but in Figure 3 it is virtually impossible to distinguish which colors belong to which functions as several different functions have very similar colors.
There is some redundant information between material and methods and the results section, which could be reduced in a few cases, notably in the section “assembly of transcripts”, where there is no need to state once more which versions of which tools were used.
In line 30 the authors claim to examine the influence of trophic mode on gene expression, where in reality this influence may be bi-directional… It may be more suitable to talk about the “relation between” these two factors.
In line 124 the authors state that cells are harvested by centrifugation, and in line 126 filters ground in liquid nitrogen. I’d assume that either filtration or centrifugations were used. Please clarify. If centrifugation was used please specify the time, temperature, and the centrifugal force. Also note that differences in this procedure may result in strong differences in gene expression…
In line 219 remove on of the “the”s; also I’d suggest removing “upper bound”.

Experimental design

The study addresses two independent questions. The first is clearly defined: which expressed genes correlate with different nutritional strategies? The second is less clearly defined, but relates to the applicability of alignment-free approaches to RNAseq data to elucidate phylogenetic relationships between organisms. Both questions are interesting, but only indirectly related. The experimental data generated allows addressing the first (although it is not optimal), but additional data would be required for the second question.

Question 1 (nutritional strategies): ideally the authors would have chosen pairs of mixtrophic/autotrophic algae from each of the 5 to 8 clades that have independently evolved heterotrophy. Instead the selection of strains seems biased towards heterotrophs. Moreover, depending on the phylogenetic tree examined, most mixotrophic strains fall into one cluster (notably in Fig 8B, where all but one mixtrophic strain form a monophyletic cluster potentially introducing a phylogenetic bias in the comparisons). It would be very useful to show the selected strains in the context of a larger and solid phylogeny of chrysophytes, so that the reader can at least get a good picture of what branches are represented.

Question 2 (alignment-free phylogenies): Knowing the exact phylogenetic position of the strains is important to evaluate the alignment free methods, yet a solid phylogenetic analysis with several established marker genes is not presented. Instead the authors cite another paper with an SSU tree, which comprises only some of the examined strains. Even for the EEF1A phylogeny no information is given about the length and quality of the alignment, making it difficult to evaluate the relevance of the resulting tree. Several of the authors have contributed to the cited Grossman 2016 paper, so they know how to do proper phylogenetic analyses. I do not quite understand why this knowledge is not applied in this paper.
The method section does not contain information about the axenic strains. How were they obtained? How was axenicity verified? What is the rational of examining both xenic and axenic strains?

Validity of the findings

Generally, the authors are careful about the conclusions they draw and do not over-interpret their data. However, I do have two concerns.

The first is the phylogenetic position of the strains – this needs to be clear to ensure that differences observed are really related to the nutritional strategy rather than to the phylogenetic position of the strains. If the mixotrophic strains cluster this at least needs to be discussed.

The second concern relates to the analysis of the 100 most expressed transcripts. This approach artificially creates differences between strains. Assume, for example that in one strain transcript A is the 99th most expressed transcript whereas in another strain it is the 101st most expressed transcript, then it will only show up in analyses first strain. Results would be less misleading if the same functions were compared across all strains.

Additional comments

This is an interesting and mainly well-written manuscript addressing an important biological and a second methodological question. Although I cannot recommend its acceptance in its current state, with proper phylogenetic analyses and some additional corrections (see above), I believe it would most likely be suitable for the Peer J. Please note that marker gene sequences could be obtained by targeted sequencing or potentially from the RNAseq data, e.g. if reads are mapped directly against conserved reference gene.

Reviewer 3 ·

Basic reporting

No comments.

Experimental design

(1) I have major concerns with how the authors obtained KEGG annotations of the transcripts. The choice of RAPSearch2 over BLAST is already questionable, as only assembled transcripts (tens of thousands of sequences) were searched, while RAPSearch2 is usually a tool used for millions of sequences, sacrificing sensitivity for speed. In this case, I don't see the need of replacing normal annotation tools such as KAAS with RAPSearch2. Second, the cutoff used in the search (log10(evalue) < 1) is way too relaxed. This basically equals no e-value cutoff. Therefore, hits between distantly related sequences of different functions could occur and may lead to incorrect, most likely false positive, KEGG annotations.

(2) While I understand the focus on the phylogenetic part of this paper is the alignment-free approach, I can't help but notice the alignment-based part needs a lot of improvement. I don't understand why the authors chose one single gene to reflect the phylogeny generated by alignment-based methods rather than generating a tree using concatenated conserved sequences like in Ciccarelli et al. 2006. One of the advantages this study have is the large amount of data they generated, which allow multiple conserved gene families to be detected (more than 200 according to the authors). A tree generated from concatenated sequences of all or some (selection of known ultra-conserved genes like those in Ciccarelli et al. 2006) would be much more convincing than a tree generated by ~200bp from a single gene.
In addition to that, when organisms are distantly related (like many of the species here, sure, they are all Chrysophytes, but many represent different classes), protein sequences are often better in reflecting phylogeny. I wonder whether the authors had tried using concatenated protein sequences to generate a tree.
Also, the authors used KO numbers as proxy for identifying orthologs, but they didn't explain what happens when more than one transcript were annotated with the same KO number.
I think, if done correctly, the alignment-based method could produce a phylogenetic tree of much higher quality that might resemble the 18S tree. If so, this might make the alignment-free trees look bad. I understand the authors want to promote the alignment-free approach, but this should not be done by making an inadequate alignment-based tree, intentionally or unintentionally. When using alignment-based tree as a benchmark, please do it properly so you have a correct benchmark. You should not be cutting corners on the alignment-based tree while at the same time listing all the difficulties associated with it.

Validity of the findings

(1) The log-scale on Fig.2 made the difference of unique KOs appear much smaller than it actually is, especially for those transcriptome with lower coverages. Since the authors' evaluation of completeness of the transcriptomes are based on the number of unique KOs (Line 237-238), this is not the proper way to show the data. Also, I recommend actually carrying out some completeness evaluation such as using BUSCO. The completeness of the transcriptomes is important in this study as it clearly affects the analysis of gene presence/absence in certain groups.
Also, line 239-240 is very confusing, it sounds like there are ~3000 genes that are conserved among all strains, which is not what the authors meant at all.

(2) The PCA analysis result is very interesting. However, the authors did not notice that the separation of heterotroph and mixotroph mainly occurred on the 2nd component, which only explains ~10% of the variance. There was no separation between blue/green dots on the 1st component at all. I noticed some of the strains close to each other on the 1st component are phylogenetically closely related, according to 18S tree at least. I suggest the authors look more closely into this. If so, this means the difference between transcriptomes are often affected more by phylogeny than trophic modes, an important point.

(3) The authors promoted the advantages of alignment-free phylogenetic approach, which is mainly speed. They proved the method is just as good if not better at species/strain level than 18S trees. Outside that, the accuracy of this method is still very questionable. I have expressed above my feeling that the alignment-based tree wasn't done optimally. If done properly, it is likely that alignment-based approach could be more accurate than alignment-free ones (Patil and Hardy, 2013), if we take 18S as the gold standard here. At this point of protistan research, I don't think the community is generating genomic/transcriptomic data at breakneck speed that the speed of data processing trumps the accuracy of the phylogeny generated. A day like that may come, but right now I think we can certainly afford a few days or even weeks to generate an more accurate phylogenetic tree. I feel the authors over-emphasized the advantages of the alignment-free approach without proving the most important part of phylogenetics study - accuracy.

Additional comments

Generally the paper is well written, but there are a few spelling errors, wrong words, or confusing sentences such as Line 156 "will we consolidated". I can't list every one of these, please check throughout the paper and correct/rewrite those.

---

## Round 0.2 · Minor Revisions

· Academic Editor

Minor Revisions

Dear Authors,

I thank you for submitting your revised manuscript. As you can see the reviewers have now accepted your paper for publication in PeerJ. Before the paper's official acceptance, please pay attention to one very minor comment, with which I also agree. This revision will not go to the reviewers if you submit a detailed answer along with the revised text. Looking forward to receive your revised manuscrpit.

Reviewer 1 ·

Basic reporting

I am pleased with the changes and the rebuttal letter that the authors provided. I believe the upon the proposed recommendations they managed to show a nice and robust work that is very interesting. I am positive to this study so I would like to accept their paper for publication.

Experimental design

Please see basic reporting

Validity of the findings

Please see basic reporting

Reviewer 2 ·

Basic reporting

see below

Experimental design

-

Validity of the findings

-

Additional comments

In this revised version of the manuscript "Comprehensive transcriptome analysis provides new insights into nutritional strategies and phylogenetic relationships of chrysophytes" Beisser et al. respond to the majority of the points raised during the review process. The additional background information provided helps to place the study in the right context and the added phylogenetic analyses and the clarifications help to correctly interpret the results from the alignment-free phylogenetic analyses and their aim. I still have two very minor comments:
Line 628 do you mean “genus” instead of “genome” ? (I missed this previously)
In line 321-325 you now state: “In order to exclude effects of medium composition and of food bacteria, we performed a likelihood ratio test comparing a full model including both nutritional strategy and media effects against a reduced model including only a nutritional strategy effect but no media effect. This analysis suggested that the medium was not a confounding factor in our analysis and did not have an influence on overall”. Given that all heterotrophic strains were cultured in inorganic basal medium with bacteria as food, and all mixo-/phototrophic strains in other media, there is no way whatsoever to separate the effects of medium and nutritional strategy. I am almost certain that if you had added a third model including only the medium effect you would not have found significant differences with the other two models either (i.e. both variables equally explain the results). In this paper it would make the most sense to simply accept that medium may be a confounding variable; this does not significantly reduce the interest of the results.
Except for this latter minor comment I recommend this paper for publication.

Reviewer 3 ·

Basic reporting

No comments.

Experimental design

Previous concerns addressed.

Validity of the findings

I like that the authors made a multigene tree according to my suggestion. Though I would like it to include more genes, I understand the difficulties they faced and the decision to cut down the number of genes. The tree is more similar to the 18S tree for several species, and the author acknowledged that the potential influence of GC content on the k-mer trees.

The authors rephrased their evaluation of the k-mer based approach properly, I think.

Additional comments

The authors addressed most of my concerns during their revision, and performed analyses according to my requests. I therefore recommend its publication.

---

## Round 0.3 · accepted · Accept

· Academic Editor

Accept

Your paper in now accepted. Thank you for considering PeerJ for publishing your research and we are looking forward to receiving your future manuscripts!